# Phytoparasitic Nematodes of *Musa* spp. with Emphasis on Sources of Genetic Resistance: A Systematic Review

**DOI:** 10.3390/plants13101299

**Published:** 2024-05-08

**Authors:** Amanda Bahiano Passos Sousa, Anelita de Jesus Rocha, Wanderley Diaciso dos Santos Oliveira, Leandro de Souza Rocha, Edson Perito Amorim

**Affiliations:** 1Department of Biological Sciences, Feira de Santana State University, Feira de Santana 44036-900, BA, Brazil; amandabahiano5@gmail.com (A.B.P.S.); diacisowanderley@hotmail.com (W.D.d.S.O.); 2Embrapa Mandioca e Fruticultura, Cruz das Almas 44380-000, BA, Brazil; anelitarocha@gmail.com (A.d.J.R.); leandro.rocha@embrapa.br (L.d.S.R.)

**Keywords:** *Musa* spp., genetic resistance, parasitic nematodes, phytonematodes

## Abstract

Bananas are a staple food that considerably contributes to both food security and income generation, especially in countries of Africa, Asia, and Central and South America. The banana plant (*Musa* spp.) is affected by various pathogens, of main concern being the plant-parasitic nematodes associated with the rhizosphere, the most important of which are *Radopholus similis* (burrowing nematode), *Helicotylenchus* sp. (spiral nematode), *Pratylenchus* sp. (root lesion nematode), and *Meloidogyne* sp. (gall nematode). Infected plants reduce their ability to absorb water and nutrients, which can lead to delayed flowering, fewer bunches, and lower fruit mass. Obtaining nematode-resistant banana cultivars through genetic improvement is an effective and sustainable option compared with chemical control with nematicides. Here, we provide the first systematic review of existing banana sources of resistance to nematodes to aid the management and control of nematodes in banana and plantain crops. Articles selected from different databases were evaluated, and searches were conducted using pre-established inclusion and exclusion criteria. We found 69 studies dealing with genetic improvement for nematode resistance in banana cultivation. Our findings revealed that sources of resistance are currently under investigation to combat the diseases caused by different nematode species in banana plants.

## 1. Introduction

Bananas and plantains (*Musa* spp.) are staple foods for more than 400 million people in the developing countries of South America, Southeast Asia, and Africa and are a key commodity in both international and local trade, particularly in Latin American and Caribbean countries [1]. These crops are ranked among the most valuable agricultural commodities in the world and are considered important for food security, as they can produce fruits throughout the year and can be cultivated in different types of environments [2]. In Africa, plantains play a crucial role in ensuring food security and income generation for more than 70 million people [3,4,5]. Although different banana cultivars are produced and consumed worldwide, the cultivars of the Cavendish subgroup are highly prominent, given their importance in the fruit export market [6]. Banana ranks second in fruit production, with global production estimated at approximately 167 million tons. They are mainly grown in Asia, Latin America, and Africa, and India and China are their largest producers for domestic consumption. In 2021, the estimated export volume was 20.5 million tons [7]. However, banana and plantain production is affected by various biotic and abiotic factors. Some notable pathogens include *Ralstonia solanacearum* (banana wilt) [2,8,9,10], *Fusarium oxysporum* f. sp. *Cubense* (Fusarium wilt) [5,11,12,13], *Mycosphaerella fijiensis* (black Sigatoka) [14,15,16] *Cosmopolites sordidus* or banana fruit borer, and nematodes (root necrosis) [17,18].

The main parasitic nematode species infecting *Musa* spp. are *Radopholus similis*, *Helicotylenchus* sp., *Meloidogyne* sp., *Pratylenchus coffeae*, and *Pratylenchus goodeyi* [19,20]. They lead to an average annual loss of 20% of banana yield worldwide [21,22]. Losses become more severe when the root system, already damaged by these parasites, experiences storms, causing the plants to topple over [23]. Additionally, the damaged roots act as a food base for fungal pathogens of the banana plant, facilitating their invasion and resulting in multiple infections, including Fusarium wilt [24,25,26].

Nematode management in banana plantations relies mainly on the use of chemical nematicides. However, in addition to being expensive, nematicides can cause environmental degradation, have the potential to leave residues on fruits, contaminate groundwater, and affect non-target organisms, and pose a risk of toxicity to applicators [27,28,29]. Cultural control measures, such as rhizome cutting followed by hot water treatment, and the use of healthy shoots or clean planting material, such as those from tissue culture, offer only temporary nematode management, as fields are usually reinfested [30,31]. Several issues, such as the frequency and method of application for biocontrol agents, particularly in field conditions, need to be addressed. In addition, the developed methods must be cost-effective, so that producers at different agricultural scales can adopt them [32].

Developing host plant resistance is an effective and economical approach to avoid yield loss due to plant-parasitic nematodes [17]. Although naturally occurring nematode resistance and tolerance have been extensively explored in many agricultural crops, nematode resistance and tolerance in banana plant have been largely overlooked, despite being present in the *Musa* gene pool [17,33,34].

Currently, banana breeding programs are focused on the introgression of various traits, including resistance and tolerance, and productivity and fruit quality [35,36], as well as the development of strategies that can surpass agricultural limitations, setting new records for yield and tolerance to biotic and abiotic stresses [37]. Banana improvement programs have made significant progress in obtaining cultivars resistant to black Sigatoka and Fusarium wilt. Based on systematic reviews on *Musa* spp. resistance to black Sigatoka and Fusarium wilt, Soares et al. [16] and Rocha et al. [5] reported that the main banana breeding programs are located in Africa, Asia, and the Americas, with around 10 genetic improvement programs for *Musa* spp. resistance worldwide [5].

Thus, various sources of resistance have already been identified and reported in *Musa* germplasm, as well as commercial cultivars resistant to the main diseases that impact banana production; however, a review of banana plant resistance to nematodes is lacking. Therefore, this systematic review was performed to determine the potential application of the current state of knowledge on *Musa* spp. resistance in genetic improvement for resistance to banana parasitic nematodes. Studies conducted between 2011 and 2024 were included in this review.

## 2. Results

### 2.1. Study Screening

Figure 1 shows the flow diagram for screening the articles analyzed in this review. Using the search *string* defined in our study, 3430 articles were found on the Google Scholar website. The first 100 pages sorted by relevance were collected, which included 1172 (31%) articles evaluated in the selection phase of this SR. The CAPES Periodicals Portal was the primary contributor for these articles, accounting for 988 articles (26%) in total. This was followed by Springer contributing to 797 articles (21%), PubMed Central with 387 articles (10%), Scopus with 312 articles (8%), Web of Science with 104 articles (3%), and MusaLit with 28 articles (1%) (Figure 1).

### 2.2. Bibliometric Analysis

A total of 3788 articles were identified in the databases, of which 482 were duplicates and 2963 were rejected in the initial selection process. In the extraction phase, 343 articles were selected, of which 69 were finally included for the SR. The selected articles were stored in an open-access digital library available at: (https://doi.org/10.5281/zenodo.10854789 (accessed on 27 March 2024)).

Most of the articles excluded at the extraction stage were related to studies dealing with other topics (96) and studies on biological control of nematodes (73), followed by studies on nematode management in banana cultivation (71) (Figure 1).

To generate a keyword co-occurrence network, we defined the minimum word frequency threshold, which enabled the visualization of all keywords occurring at and above this frequency. The algorithm of the VOSviewer software version 1.6.19 generated a network based on the association method with many significant clusters. Each node in Figure 2 represents a keyword, and the radius of a node is related to the frequency of occurrence of the keyword in each article. According to the number of nodes, the central brown cluster comprises the most frequently used concepts and those that have received attention from most researchers, with the word “banana” in the center as the cluster’s main theme. As shown in the map in Figure 2A, “nematode”, “*musa*”, “*Radopholus similis*”, and “*Meloidogyne incognita*”, are among the ten most common keywords in each of the other clusters.

The graph presented in Figure 2B demonstrates the collaboration network among countries based on co-authorship. The network shows that for the topic of our study, cooperative relationships existed among nearby countries and that the clusters formed appeared to be related to the geographical proximity of each country. The largest cluster, in purple, had the largest collaboration network with other countries and was primarily made up of India with closer cooperation with the Philippines and France. The next largest clusters are shown in brown, represented by Belgium, and green, represented by Uganda. Belgium had a greater cooperation with the Philippines and African countries, such as Uganda, Nigeria, South Africa, and Kenya. The fourth largest cluster is shown in orange, represented by Brazil, whose main collaboration networks were with the United States, Costa Rica, and Colombia, which are the closest countries geographically. Other clusters with extensive collaboration networks included the United States, China, and France (Figure 2B).

We generated a network diagram for the main journals as shown in Figure 3A. The top three journals with the highest number of articles published on the subject of our study were Nematology, Nematropica, and Indian Journal of Nematology, which were distributed in five different clusters determined by the central colors. There was a greater cooperation between the journal Nematology and the journals Nematropica, European Journal of Plant Pathology, and Indian Journal of Nematology. Figure 3B also shows the relationship graph of the cooperation and co-occurrence network of authors involved in studies on banana resistance to parasitic nematodes. Seenivasan N, Das Sukhen, and Roderick Hugh were the authors who appeared in the highlighted groups, indicating that they have published most of the articles included in our study or collaborated most with other researchers. In addition, our data showed that not many scholars were engaged in this area of banana research and no cooperative relationship existed among them.

### 2.3. Places of Origin

Figure 4 shows a map with the frequency of articles by country, represented by a scale with a color grid that goes from yellow, which represents the minimum frequency (1%), to red, which represents the maximum frequency (54%). Thus, among 69 studies, 45.6% were conducted in India, 13.2% in Brazil, 8.8% in Uganda, 4.4% in Malaysia and Egypt, 2.9% in the Philippines and Costa Rica, and the remaining in countries such as Germany, Costa Rica, Ivory Coast, Belgium, Cameroon, and France.

Pie charts integrated into Figure 4 show the general frequency of nematodes studied and their distribution in the countries. Overall, *R. similis* was the most widely studied nematode, specifically in the countries of Africa, Europe, and Asia. In American countries, *Meloidogyne* spp. and *Helicotylenchus* spp. were the most frequently occurring nematodes (Figure 4). The countries with the greatest diversity of species studied were India with 11 species reported, Brazil with 8, and Malaysia with 7, in addition to other species reported at a lower frequency (Figure 4).

*R. similis* had the greatest frequency and distribution among the countries that have conducted studies on nematode resistance in banana (20.7%), followed by *P. coffeae* and *Meloidogyne* spp. (both 10.6%), *Helicotylenchus multicinctus* (9.6%), and *M. incognita* (8.1%). *Rotylenchulus reniformis*, *P. goodeyi*, *Helicotylenchus dihystera*, and *Meloidogyne javanica* appeared in 4%, 2.5%, 2%, and 1.5% of the articles. Additionally, some studies have reported other genera without identifying the species, such as *Helicotylenchus* and *Pratylenchus* (both 6.6%), as well as *Radopholus* (2%) and *Rotylenchulus* (1.5%). Other nematodes of lesser importance were also identified in 13.6% of the articles, mainly in those that dealt with population surveys in the field. The frequency considered the evaluation of more than one nematode per article (Figure 4).

The main nematodes studied in India were *P. coffeae* (n = 14), *R. similis* (n = 12), and *M. incognita* (n = 8). In Uganda, they were *R. similis* (n = 7), *H. multicinctus* (n = 6), and *Meloidogyne* spp. (n = 5), considering more than one nematode per article. In Brazil, the most studied nematode was *R. similis* (n = 5). In Costa Rica, *R. similis* (n = 2), *Helicotylenchus* spp. (n = 2), *Pratylenchus* spp. (n = 1), and *Meloidogyne* spp. (*n* = 1) were evaluated together in two different studies. Côte d’Ivoire and France studies have reported *R. similis* (n = 1) and *P. coffeae* (n = 1). Countries such as Egypt and Malaysia have evaluated nematodes such as *M. incognita* (n = 3) and *Rotylenchulus reniformis* (n = 2). *R. similis* and *M. incognita* have been less frequently studied by other countries.

In the selected articles, eight breeding programs from different countries were found, containing information on genetic improvement for banana resistance to parasitic nematodes: Brazilian Agricultural Research Corporation (Embrapa) in Brazil; Improvement Program Department of Fruticulture, National Research Centre for Banana (ICAR), and Horticultural College and Research Institute (HCRI) in India; International Institute of Tropical Agriculture (IITA) in Uganda; Institut de Recherche pour le Développement (IRD) and Centre de Coopération Internationale en Recherche Agronomique pour le Développement (CIRAD) in France; Center National for Research Agronomique (CNRA) in Ivory Coast; and Agriculture Research Center (ARC) in Egypt.

Of the 69 articles selected, 24 included population surveys of field nematodes in different regions. For this evaluation, studies that specified the nematode species, not just the genus, were separated out, thus a final restriction to 11 countries. An analysis was inserted based on a heat map, in which the population densities of different nematode species were expressed in colors ranging from red (high density), orange (medium density), and yellow (low density) (Figure 5). This analysis was associated with the world location map with the total population densities of each nematode by country (Figure 5).

Although most of the studies have been carried out in India and larger numbers of nematode species have been evaluated in India (Figure 4), the population density of these species in banana plantations did not appear to be higher in India than in other countries. Colombia was the country with the highest population density of a single nematode, *R. similis* (19,489); in Malaysia, studies have reported higher population densities of *M. incognita* (3850), *R. reniformis* (2450), *H. multicinctus* (2450), and *H. dihystera* (2400). Uganda had a higher population density of *P. goodeyi* (8215) and intermediate densities of *H. multicinctus* (2008) and *R. similis* (1739) (Figure 5). In Zimbabwe and Costa Rica, the highest densities were for the nematode *R. similis* (3429 and 4766, respectively). In this analysis, the frequency considered the total population density across all areas evaluated per article.

### 2.4. Evaluation Tools and Methods

Regarding the environment in which the studies were conducted, most studies were performed in the field (44%), including population surveys in different areas, followed by greenhouses, both of which accounted for 17% of the articles. Additionally, 6% of the studies included screenhouse experiments and 2% included in vitro and laboratory experiments. However, 9% of the articles did not specify where the experiments were conducted (Figure 6A).

Regarding the main methods used to characterize nematode-resistant plants, nematode density/reproduction factor analysis was addressed in 32% of the selected articles, with the highest frequency for evaluating banana resistance to nematodes, followed by symptomatology (16.7%), morphological/molecular characterization (16%), enzymatic activities and phenolic compounds (11.3%), hybridization (8%), symptomatology/agronomic characteristics (6%), gene expression analysis (3.3%), transgenics (2.7%), selection assisted by molecular markers (1.3%), and others (2.7%). The frequency was measured considering that more than one tool was used per article (Figure 6B).

Figure 7 shows the main evaluations of the symptoms caused by different nematode species on banana trees, with some authors using rating scales. The scale, previously used by Pinochet [38], identifies susceptibility, tolerance, or resistance, and was used in six studies (27%) for *P. coffeae*, *H. multicinctus*, and *R. similis* (Table 1). There were papers that used different indices to assess root lesions and/or necrosis (19%) and the extent of root damage (15%). The articles that assessed damage caused by *M. incognita* used a root gall index scale (15%). Additionally, 8% of the articles assessed the index of necrosis in the rhizome and 4% assessed the number of functional roots. Other methods were also used to assess symptoms (12%). The frequency considered that more than one method was used per article.

The methods for extracting nematodes in roots were also demonstrated in most of the selected articles (Table 2). Some articles only provided the reference of the protocol followed, without giving details, and other authors did not specify how they carried out the counting. To assess the population of *R. similis*, *P. coffeae*, and *Helicotylenchus* spp., the most commonly used technique was as follows: roots were first macerated in a blender for 10 s thrice with an interval of 5 s and then sieved [17,21,27,39,40]. Araya and De Waele [41] used a method where the roots were macerated in a blender for 10 s each at low and high speed, followed by sieving. Tripathi et al. [23] and Sankar et al. [42] macerated the samples in a blender for 10 s. Another method used for nematode extraction consisted of placing root samples in polypropylene bags and submerging them in 100 mL of 1% H_2_O_2_ solution and incubating them at room temperature for 7 days in the dark; thereafter, the nematodes were collected and counted using a microscope [43,44]. To assess the population of *Meloidogyne* spp., the most commonly used techniques were staining the roots with lactophenol acid fuchsin [29,45,46,47] and acid fuchsin in acetic acid [48] without macerating the root tissues. The selected articles used microscopy counting.

Among the tools used to analyze and characterize nematode-resistant plants, the frequency of articles using reverse transcription PCR (RT-qPCR) and PCR analysis was 46.5%; histology/histochemistry, tissue culture, and chromatography, accounted for 9.3% of the works evaluated; banana transcriptome, bioinformatics, and phylogenetics, accounting for 7% of the articles; and proteomics accounting for 4.7% (Figure 8A). Germplasm selection was mainly carried out through symptom evaluation, which takes into account the study environment. Consequently, greenhouse symptomology appeared in 28.6% of the publications, followed by field symptomatology (14.3%), greenhouse population evaluation (11.4%), and field agronomic characteristics (8.6%). In addition to these, there were also other germplasm selection techniques with less frequency (Figure 8B).

### 2.5. Sources of Resistance

The most frequently reported known sources of resistance to nematodes were the genotypes Pisang Lilin (AA), Yangambi km5 (AAA), Karthobiumtham (ABB), and Anaikomban (AA). In addition to these, other cultivars were also used as sources of resistance to nematodes, such as the hybrids FHIA-21, FHIA-23, and FB920 (Figure 9A). Of the reported sources of resistance, the AA diploid genome predominately appeared, followed by the AAA and AAB triploids (Figure 9B).

Many cultivars were used to confirm or test resistance to different nematode species. Table 3 shows the genotypes that were resistant, moderately resistant, or tolerant to each nematode. Of the genotypes reported, 41% were AA diploids, 23% were AAA triploids, 15% were AABB tetraploids, 14% were AAB triploids, 5% were AAAB tetraploids, and 3% were AB diploids (Figure 10).

### 2.6. Gene Expression Analysis

Most of the articles that evaluated gene expression studied the nematode *P. coffeae* [25,67,70,72]. One study reported banana resistance to *M. incognita* [76]. Table 4 shows the candidate genes that are differentially expressed and potentially involved in defense responses to different nematodes identified in the selected articles.

## 3. Discussion

### 3.1. Study Screening and Places of Origin

The selection of articles in this review were restricted to genetic improvement, following the protocol developed. In total, 69 articles were included in the data synthesis and analysis. As shown in Figure 1, various articles involving nematodes and *Musa* spp. were identified, but they did not focus on genetic improvement or answer the SR questions (n = 96).

Most of the articles evaluated in this SR regarding the genetic improvement of banana plants for nematode resistance were conducted in India (45.6%), indicating that this country occupies the largest area of banana cultivation and production in the world [77]. In addition, among banana pathogens, plant-parasitic nematodes play a crucial role in crop loss by decreasing productivity [78], which can be proven by the diversity of nematode species found in India compared with other countries (Figure 4). Brazil and Uganda also contributed to the genetic improvement of *Musa* spp., accounting for 13.2% and 8.8% of the selected articles, respectively. These countries are home to important research institutions working on banana improvement, such as Embrapa in Brazil, which ranks fourth among banana producers [79], and IITA in Uganda.

Among these results, some studies conducted under controlled conditions, involved more than one nematode (n = 7). These studies compared the responses of cultivars to the genus or species of nematode used. For instance, Araya and De Waele [41] found that the horizontal and vertical distributions of *R. similis* in the root system of Valery, Gros Michel, and FHIA-18 were considerably similar. However, the distributions of *Helicotylenchus* spp. and *Meloidogyne* spp. and the total number of nematodes varied slightly among the genotypes. Vawa et al. [68] found that the hybrid FHIA 21 was resistant to *R. similis* and susceptible to *P. coffeae*. *P. coffeae* was the second most widely studied nematode and exhibits symptoms similar to those caused by *R. similis* [64]. The greater damage capacity of *P. coffeae* may be related to the parasite’s ability to colonize all cellular compartments of the root system [68,80]. In addition, the biological cycle of *P. coffeae* is shorter than that of *R. similis*, and *P. coffeae* more rapidly spreads than *R. similis* [68].

We found 24 studies that performed population surveys in areas with banana plantations infested with nematodes. The majority of these research was conducted in India (n = 7) and Brazil (n = 4). Except for the USA and Tanzania, studies from all other countries have reported the nematode *R. similis* in the rhizosphere, which is considered as one of the main causes of banana production losses in the world.

Famina et al. [81] investigated the nematodes present in the rhizosphere of banana trees in the district of Malappuram, India, and found a total of 10 species, namely *R. similis, H. multicinctus*, *H. dihystera*, *P. coffeae*, *Hopolaimus galeatus*, *R. reniformis*, *M. incognita*, *Heterodera glycines*, *Hemicycliophora arenaria*, and *Criconemella* sp. Among these, *Helicotylenchus* sp. occurred most frequently, followed by *R. similis*.

In contrast, Odala et al. [78] found that the presence of *R. similis* in the region of Attappady, India, was less common compared with other nematode species, such as *Meloidogyne* spp., *Pratilenchus* spp., and *Rotylenchulus* spp. In addition, the cultivars evaluated in the selected areas exhibited variations in their response to nematode attacks, with the Nendran cultivar being the most susceptible to phytonematodes.

The pattern of nematode infestation in banana plants found in natural habitats indicates that the occurrence and predominance of a particular species in one country may not be similar in neighboring countries [82]. However, diversity and population survey studies indicate that the genetic basis for nematode resistance in banana accessions requires further investigation [83], for which it is important to focus on the management of phytonematodes that parasitize the crop [84].

### 3.2. Evaluation Tools and Methods

Since nematode population directly damages the root system, causing lesions, assessing damage to the root and rhizome is of great importance [17].

The primary approach for assessing the symptoms in these studies was through rating scales, which can classify the lesions. The scale, previously used by Pinochet [38], identifies susceptibility, tolerance, or resistance, and was used to evaluate the reaction of *Musa* to *P. coffeae* [27], *H. multicinctus* [21,39], and *R. similis* [17,22,40,42]; this technique evaluates the root lesion index and the degree of the rhizomes.

Rocha et al. [26] and Kosma et al. [85] evaluated the rhizomes of plants infected with *R. similis* based on the percentage of necrosis using the scale established by Bridge [86]: necrosis in <25% of the rhizome was considered mild, from 25% to 50% moderate, from 51% to 75% severe, and >75% very severe. In addition, Rocha et al. [26] assessed the number of functional roots. Ramesh Kumar et al. [65] evaluated the root lesions caused by *R. similis* and *P. coffeae* using the index described by Sundararaju [87] and the root necrosis index described by Carlier et al. [88]. The highest index (5) was recorded in susceptible cultivars, while the lowest index (1) was recorded in the resistant cultivar Pisang Lilin. The mutants from both groups, Ro Im V4 6-1-1 and Si Im V4 10-5-3, had a relatively lower index (2). Based on the percentage of root necrosis, the Im V4 6-1-1 and Si Im V4 10-5-3 mutants were considered resistant, while the Ro Im V4-6-2-1 mutant was moderately resistant [65].

Herradura et al. [61] evaluated the root necrosis index for *R. similis* following the method of Speijer and De Waele [29]. The extent of root damage was described by the following scores: 1 if the roots were all healthy, 2 if most roots were healthy, 3 if most roots were dead, and 4 if all the roots were dead. This scoring has been used for evaluating damage caused by *R. similis* [42,69] as well as *H. multicinctus* [21,39].

In the study by Speijer and De Waele [29], the evaluation of the data obtained during nematode resistance/tolerance screening was based on a combination of nematode reproduction data and host plant response data. This included at least the number of nematodes in the roots, percentage of dead roots, and root necrosis index by preference, which were taken at different stages of plant growth. The combination of these data can give a reliable indication of whether the genotype is resistant or susceptible, tolerant, or sensitive.

In Table 2, the nematode extraction methods in roots were demonstrated. However, according to Abd-Elgawad [89], the conventional nematode extraction methods need to be optimized because they present some issues related to evaluating their populations, distribution patterns, and interactions with many other factors in the context of integrated pest management. For instance, sieving and centrifugation using a sucrose gradient can extract and quantify both dead and live nematodes, unlike the Baermann funnel method, which can only extract live nematodes [89]. This statement highlights that the choice of method can lead to errors in nematode population assessments regarding plant damage.

Transgenic approaches (2.7%) have also been used, although only in few studies related to genetic improvement. Most of the transformation protocols employed were based on using cystatin and peptide to provide single or double defense against nematodes. One of the approaches used for transgenic resistance to nematodes involves interrupting their feeding. Cysteine proteinases are the main digestive enzymes of many nematodes, and small protein inhibitors (cystatins) from plants have mediated nematode resistance when expressed in various crops [23]. In contrast, defense based on plant roots secretions, such as synthetic peptides, disrupts nematode chemoreception and interferes with perception of host plant location [43,44]. *R. similis* and *H. multicinctus*, nematodes that were introduced in the study by Tripathi et al. [23], were unable to maintain their density in the root system of growing transgenic bananas, precisely because cystatin and the peptide provide a high level of resistance in bananas.

A total of 8% of the articles used hybrids that had been screened to detect possible genotypes resistant or tolerant to various nematode species. *R. similis, P. coffeae, M. incognita*, and *H. multicinctus* were the most commonly tested nematodes in these studies. Two hybrids, H516 (AAA) and H531 (AAB), proved to be resistant to the four nematode species mentioned above [21,27,45,75]. These hybrids were developed from genotypes considered sources of nematode resistance. H516 (AA) is a hybrid of Anaikomban x Pisang Lilin and H531 (AAB) is a hybrid of Poovan x Pisang Lilin. In addition to these, some other hybrids have been shown to be tolerant to more than one species of nematode. Traditional genetic improvements come from potential and improved diploids [21]. Tools such as marker-assisted selection, double haploids, and genomic selection can further accelerate population improvement at the diploid level [90]. Wild diploid bananas of *M. acuminata,* such as Calcutta 4, and other diploid cultivars, have AA genomes and may harbor important sources of resistance genes for the genetic improvement of triploid cultivars [14,16].

Although only few studies in this SR were related to molecular marker-assisted selection (1.3%), this tool plays an important role in the genetic improvement of any crop, allowing the analysis of genetic diversity, construction of genetic linkage maps, and development of QTLs for alleles linked to specific characteristics, such as resistance to biotic and abiotic stresses, fruit quality, and parthenocarpy [36]. For example, Afifi and Zawam [48] aimed to select nematode-resistant banana plants through induced mutation. They used ISSR molecular markers to observe the genetic similarity among the banana cultivars tested and found that this similarity was substantially low, which can be attributed to its mutagenesis effect. Backiyarani et al. [36] developed a database (MusatransSSRDB) that provides information on in silico polymorphic SSRs between contrasting banana cultivars for each stress and within the stress, thus facilitating the retrieval of results on cultivars, stresses, and polymorphism. According to the authors, the information contained in MusatransSSRDB makes it easier for banana breeders to select SSR primers based on specific objectives, including stresses caused by the nematode *P. coffeae*.

In the articles that evaluated enzyme activities and phenolic compounds (11.3%), all the genotypes considered resistant or tolerant had high levels of phenols, lignin, peroxidase, polyphenol oxidase, and phenylalanine ammonia lyase. Enzyme activity is one of the important tools for confirming resistance to nematodes. When a pathogen infects the host tissue, a small number of specific genes are induced to produce mRNAs that enable the synthesis of a similar number of specific proteins [27,91]. Lignin and phenol are synthesized via the phenylpropanoid pathways, which confer resistance against nematode attacks [22].

### 3.3. Sources of Resistance

Considering that banana genotypes susceptible to nematodes are predominantly consumed globally, crop improvement to develop new cultivars that offer high quality, high yield, and resistance to biotic stresses is of great importance for the global banana industry [76].

In the present study, diploid AA genomes occurred frequently, both in the already known sources of resistance with 35.3% incidence (Figure 9B) and in those tested with 41% incidence (Figure 10). This shows that the sources of resistance related to nematodes are mostly made up of diploids. This suggests that wild diploid *Musa acuminata* bananas may harbor important sources of resistance genes for the genetic improvement of triploid cultivars [14,16]. However, among the genotypes mostly used in the selected publications, in addition to the diploid Pisang Lilin, which has served as a parent for the generation of hybrids with potential resistance or tolerance, the resistant triploid Yangambi km5 also stood out.

In addition, many studies have tested nematode resistance in various hybrids. A triploid synthetic hybrid (FB920), with tolerance to yellow and black Sigatoka and partial resistance to nematodes, was tested by Quénéhervé et al. [66], who observed that tolerance to the nematodes *R. similis* and *P. coffeae*, in a field trial, was greater than for the Cavendish cultivars. However, FB920 produces small bunches and tall plants, because of which it may not be suitable for export. Nonetheless, according to the authors, these characteristics should not hinder its cultivation by small-scale producers.

### 3.4. Gene Expression Analysis

The vast majority of banana and plantain cultivars are interspecific hybrids of *M. acuminata* and *M. balbisiana* [92], and the sequence similarity is partially attributed to their genomic compositions [73]. In this review, 11.6% of the articles used the banana genome as an analysis tool. Of these, five were based on gene expression analysis, which used the RT-qPCR/PCR tool. RT-qPCR/PCR has been used by 46.5% of the selected articles in this review, involving molecular marker-assisted selection (1.3%) and transgenics (2.7%). Bioinformatics (7%) and histology/histochemistry (9.3%) also corroborated the results in some of these studies.

These findings imply that nematode invasion triggers multiple signaling pathways, both through tissue damage caused by the invasion of these pathogens and through the recognition of nematode elicitors by R genes [25].

Backiyarani et al. [25] found that the hydrolytic enzyme 1,3-glucanase was upregulated in resistant and susceptible cultivars during infection by *P. coffeae* but, on the 7th day after inoculation (DAI), the level of glucanase was abundant and twice as high in the resistant cultivar compared with the susceptible cultivar. A similar type of expression profile was observed for the peroxidase gene, in which the expression level of resistant cultivars was twice as high as that of susceptible ones and reached the maximum expression level on the 6th DAI [25].

The polyphenol oxidase (PPO) transcript was found in resistant and susceptible cultivars from uninoculated root samples, indicating constitutive expression of the PPO gene, while PPO mRNA levels were higher in roots of resistant plants compared with susceptible plants [70]. The differences in PPO transcript level between resistant and susceptible banana roots led to the consideration of potential PPO effects in *P. coffeae* [70]. PPOs are induced in response to biotic and abiotic stresses in plants and have been applied to various functional processes, including plant defense and regulation of plastidic oxygen levels and the phenylpropanoid pathway [93].

Many genes described in these studies represent interesting candidates for further analysis of host defense or susceptibility function. Most studies related to gene expression have tested the host response to *P. coffeae* [25,67,70,72]. The invasion caused by this parasite triggers multiple signaling pathways through wounds caused by the penetrating action of the stylet and the recognition of the presence of nematodes [25,94]. In addition, banana plants can recognize it by detecting compounds in the cuticle or secretions made by the nematode or both mediated by the triggered defense response genes [25].

In addition to evaluating gene expressions by RT qPCR/PCR, two studies used proteomics as an analysis tool for *M. incognita*. When inoculation with this nematode was tested, the PR10 protein acted against the invasion of this pathogen in Grande Naine, showing ribonuclease activity or β-1,3-glucanase activity. This protein showed a significant decrease in abundance in *M. incognita in* the root tissues of Grande Naine at 60 DAI. PR10 activity has been associated with the plant’s defense response, either through a direct antagonistic interaction with pathogens that invade cells or by increasing the plant’s immunity through the induction of programmed cell death around an infection site [95]. Al-ldrus et al. [96] observed that 114 banana root proteins showed significant changes when inoculated with *M. incognita* at 30 and 60 DAI. The study revealed that these changes affected proteins mainly involved in fundamental biological processes, organization of cellular components, and stress responses.

### 3.5. Final Considerations, Limitations, and Future Perspectives

There is a need to develop banana cultivars that are resistant to parasitic nematodes. Therefore, future scientific objectives should prioritize increasing the benefits offered by improved banana plants with durable resistance characteristics to this stress, preferably in cultivable varieties.

This SR was highly specific to genetic improvement for the resistance of *Musa* spp. to plant-parasitic nematodes. Hence, the number of selected articles was limited to 69. We found that, over the last 13 years, some researchers have endeavored to demonstrate methods of genetic improvement to overcome attacks on *Musa* spp. by phytopathogenic nematodes. Much of this work still needs to be improved, and further studies need to be conducted to identify a safe and efficient source of resistance against nematodes. However, important tools and resources related to genetic improvement have been developed in recent years to better understand the interaction between nematodes and banana plants, including hybridization, transgenics, enzymatic data, proteomics, gene identification, and host defense response. This has enhanced our understanding of how genotypes respond to parasitic attacks and their mechanisms of defense, indicating potential for the development of resistant commercial cultivars.

Our findings revealed that the genotypes considered as resistance sources have different degrees of resistance or susceptibility to different nematode species, or even to the same species from different geographical locations. This variability is the biggest challenge for breeding programs. The genotypes Yangambi km5 and Pisang Lilin are the most widely investigated resistance sources, mainly through studies in India, and these genotypes could be targeted by other breeding programs in future studies. Work related to hybridization has also shown great potential in developing resistance against different nematode species. Selected hybrids, such as H516 and H531, showed resistance against the four most important nematode species in banana cultivation: *R. similis*, *P. coffeae*, *M. incognita*, and *H. multicinctus*. Future breeding studies need to be improved to confirm this resistance.

We did not identify a method that has been studied more extensively, and there is a growing push for new, precise, and efficient technologies. However, all the methods mentioned in the selected studies contribute to identifying cultivars with potential resistance. The functional characterization of genes, for example, can facilitate the development of new breeding strategies.

## 4. Materials and Methods

This systematic review (SR) was performed in line with the Preferred Reporting Items for Systematic Reviews and Meta-Analyses (PRISMA) guidelines, using the open-access software StArt (State of the Art through Systematic Review) v. 3.3 Beta 03. Three stages were defined for the construction of the SR: planning, execution, and summarization.

In the planning stage, we developed a protocol for the review process, in which the title, objective, keywords, research questions, research sources, and inclusion/exclusion criteria of the articles were defined for the selection and extraction of relevant papers. This review protocol was registered in the database (https://doi.org/10.5281/zenodo.11047703 (accessed on 27 March 2024)). The main research question was structured according to the five-component strategy PICOS (Population, Intervention, Comparison, Outcome, and Study design) [97] (Table 5).

Thus, the question established in the SR was as follows: “What knowledge has been generated in the last 13 years regarding the genetic improvement of *Musa* spp., which has potential applications in resistance to parasitic nematodes in banana?” The secondary research questions are listed in Table 6.

For question 3, if the article did not mention the study location, the search criteria within the article were standardized to the first author’s mailing address to determine the country of origin for the studies.

The execution stage involved three phases: search, selection, and extraction. The electronic searches were conducted using a search *string*, defined with the following terms: (“*Musa* spp.” OR bananas OR plantains) AND (nematodes OR “plant parasitic nematodes” OR phytonematodes). The following databases were used: Web of Science, PubMed Central, Springer, CAPES Periodicals Portal, and Scopus. We also used Google Scholar, as well as MusaLit, a bibliographic database with over 18,000 references on bananas. Because MusaLit cannot identify long *strings*, the search *string* for this database was altered to (banana AND nematode), and the filtering method available in this database was used; these limitations imposed some restrictions on the results. The results were imported from each database into the BibTeX, MEDILINE, or RIS formats and then imported into the StArt software v. 3.3 Beta 03.

In the selection phase, the articles were only evaluated by reading the title, abstract, or keywords. We excluded duplicate articles, as well as articles that were not in line with the objectives of our work, such as review articles, non-English language articles, theses, dissertations, manuals, articles published before 2011, book chapters, and articles published in conference proceedings.

In the extraction phase, the articles chosen in the selection phase were read in full, and of these, only (I) the articles that answered the questions in the study protocol (Table 6) were included. The exclusion criteria for this phase were as follows: (E) first reports, (E) chemical control, (E) biological control, (E) management, (E) pathogenicity, and (E) other topics.

The summarization stage involved synthesizing all the answers to the questions proposed in Table 6. The number of articles per answer to each question was quantified, and the values were expressed as a percentage in each graph or summarized in tables. Microsoft Excel and the *ggplot2* and *dplyr* packages in the R statistical software were used to organize the data and construct the graphs. A bibliometric analysis using the VOSviewer software version 1.6.19 was inserted to check the networks of interactions among keywords, among countries, among most-publishing journals, and among authors and co-citations [98].

To avoid any risk of bias, the prism checklist was drawn up in accordance with the PRISMA standards [99]. This document is available for download at the following link: (https://doi.org/10.5281/zenodo.11047688 (accessed on 27 March 2024)).

## Figures and Tables

**Figure 1 plants-13-01299-f001:**
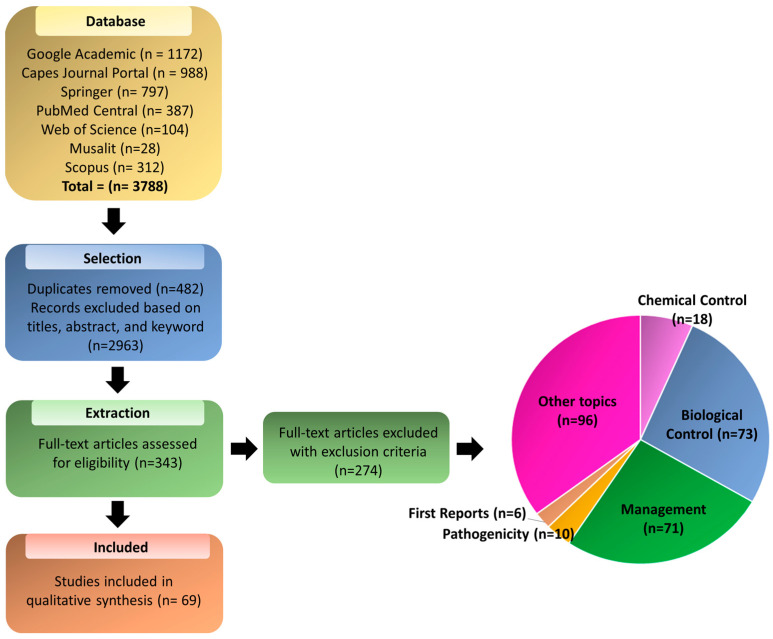
PRISMA flowchart for the screening process of the articles selected in this review.

**Figure 2 plants-13-01299-f002:**
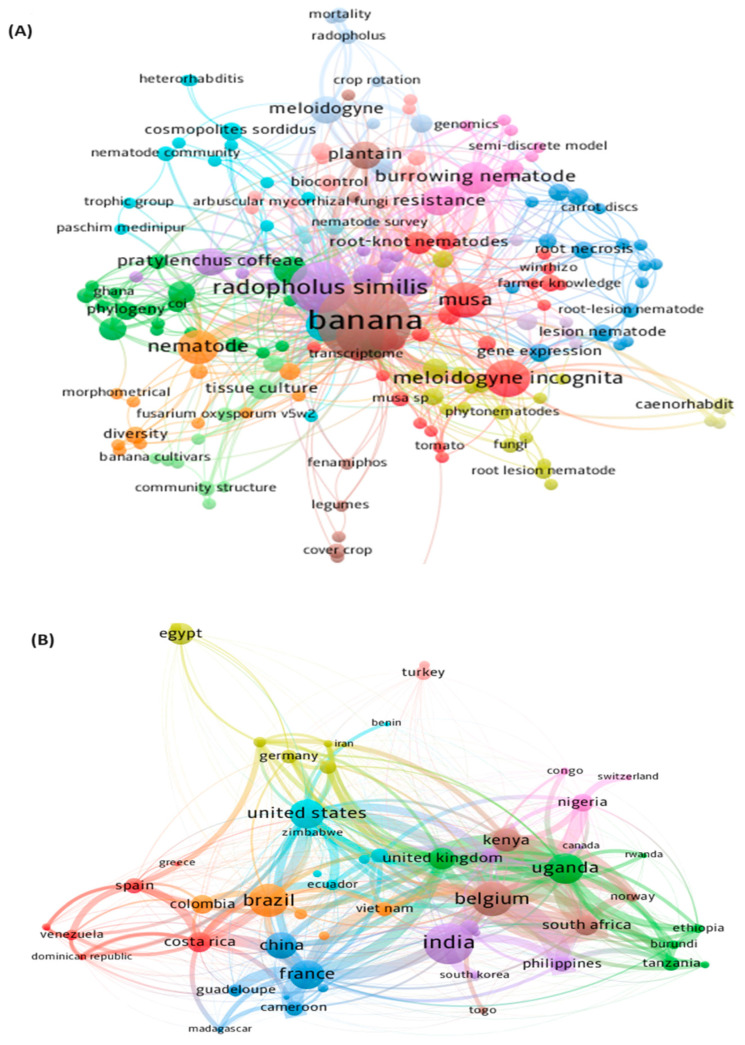
(**A**) Keyword co-occurrence networks and (**B**) collaboration networks between countries according to co-authorship. The size of the circle represents the frequency of occurrences of keywords or countries, with larger circles indicating higher occurrences, and the thickness of the lines reflects the strength of the collaboration, with thicker lines indicating closer collaboration networks.

**Figure 3 plants-13-01299-f003:**
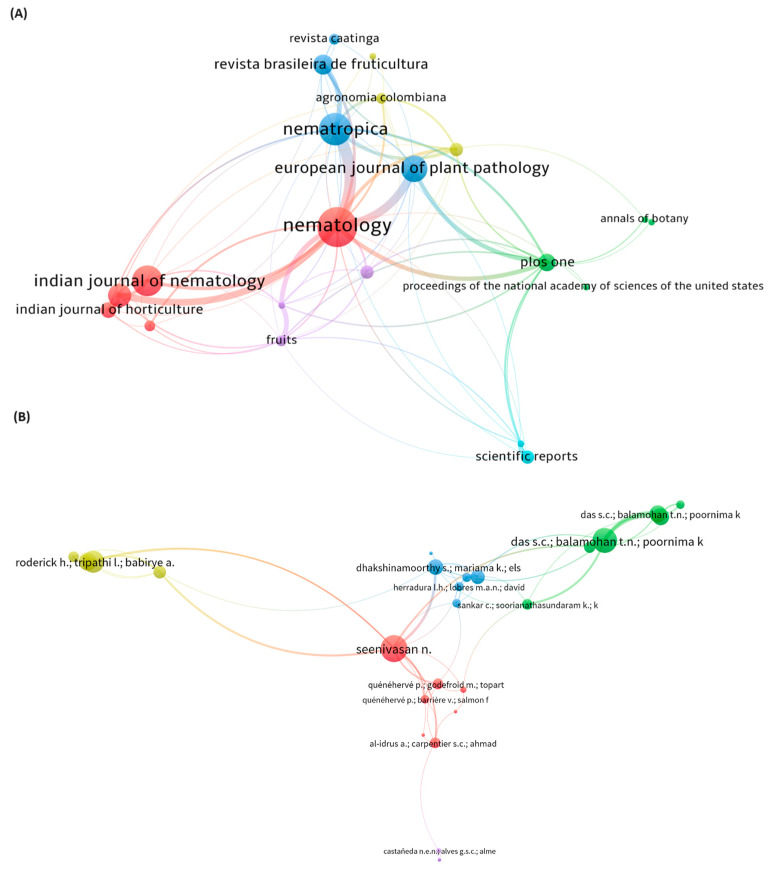
Bibliographic coupling of the main research sources (**A**). The size of the circle is directly proportional to the number of publications, and the thickness of the line connecting the circles is proportional to the collaboration among the journals. Networks of author collaborations on banana resistance to parasitic nematodes (**B**). The size of the circle is directly proportional to the number of articles published by the author, and the thickness of the lines is directly proportional to the closeness of the collaboration among the authors.

**Figure 4 plants-13-01299-f004:**
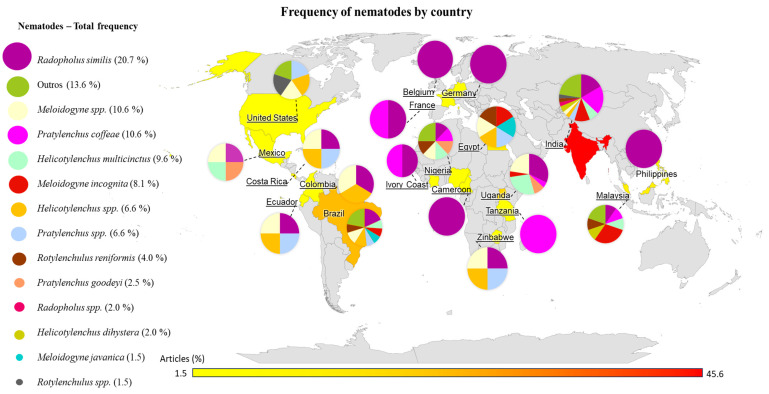
Frequency of articles on genetic improvement of *Musa* spp. to parasitic nematodes published in the last 13 years in different countries and species of nematodes studied per country, considering more than one nematode per publication. The map was plotted in R, using the maps, ggmap, geosphere, Eurostat, GADMTools, country code, and ggplot2 packages. lat: latitude; long: longitude.

**Figure 5 plants-13-01299-f005:**
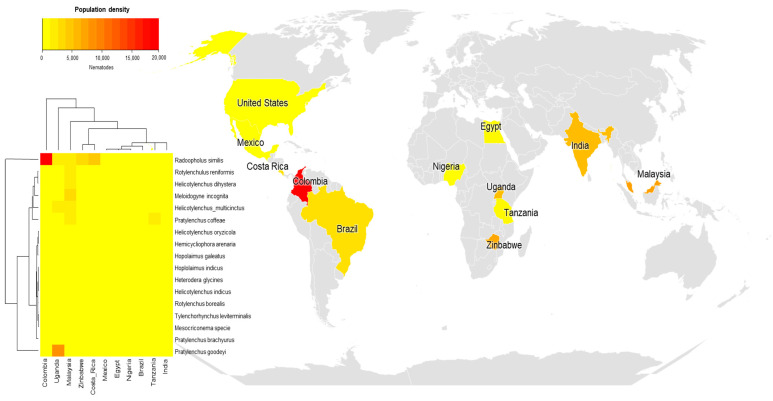
Population densities of banana parasitic nematode species reported in different countries in articles published in the last 13 years.

**Figure 6 plants-13-01299-f006:**
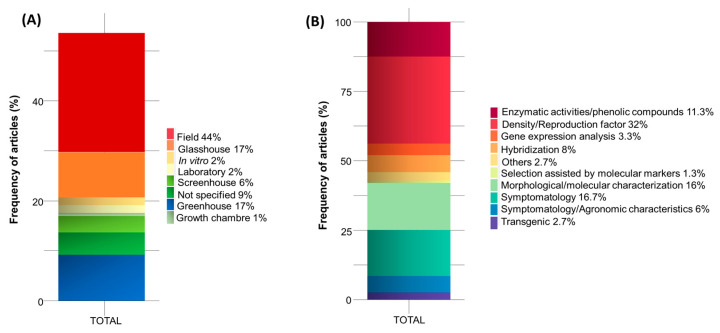
Stacked bar chart of the frequency of articles on nematode resistance to banana plants carried out in the last 13 years (**A**). Stacked bar graph of the frequency of articles with methods used to obtain or characterize nematode-resistant *Musa* spp. plants carried out in the last 13 years (**B**).

**Figure 7 plants-13-01299-f007:**
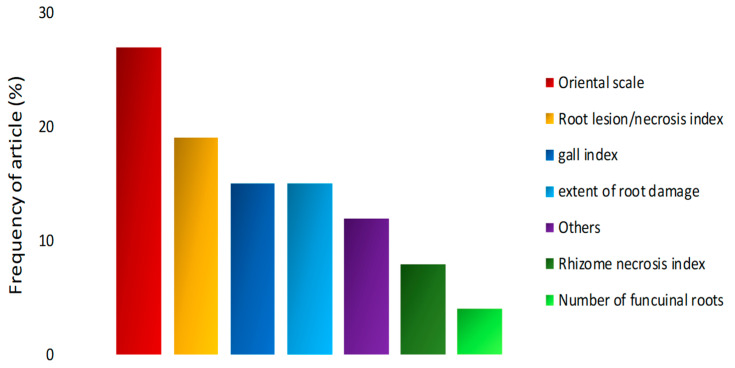
Methods for assessing symptoms caused by nematodes in banana plants in articles selected over the last 13 years.

**Figure 8 plants-13-01299-f008:**
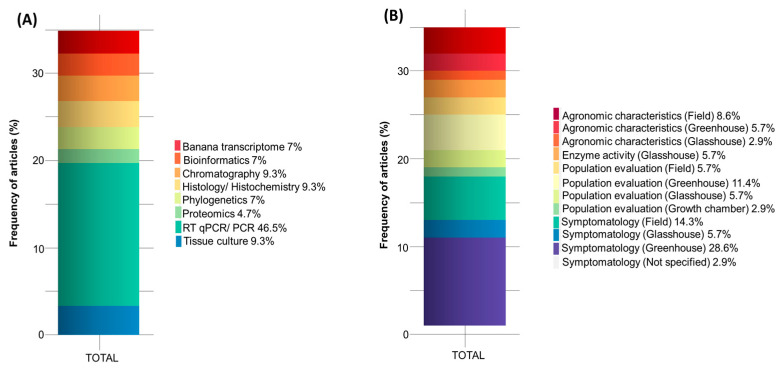
Stacked bar chart of article frequency with tools used in banana breeding studies for nematodes carried out over the last 13 years (**A**). Stacked bar chart of article frequency representing banana germplasm selection over the last 13 years (**B**).

**Figure 9 plants-13-01299-f009:**
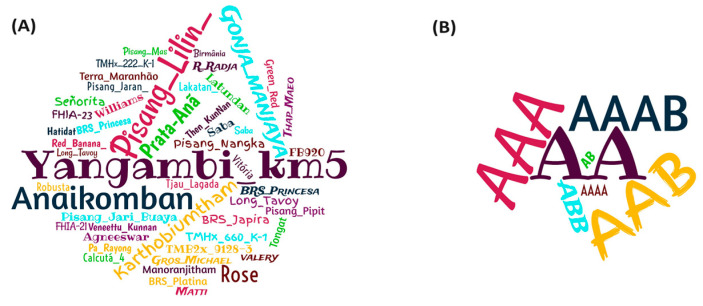
Word cloud generated from genotypes used as sources of resistance in articles selected to compose a systematic review on the resistance of *Musa* spp. to nematodes (**A**). Frequency of genomes associated with sources of nematode resistance in banana breeding studies carried out over the last 13 years (**B**).

**Figure 10 plants-13-01299-f010:**
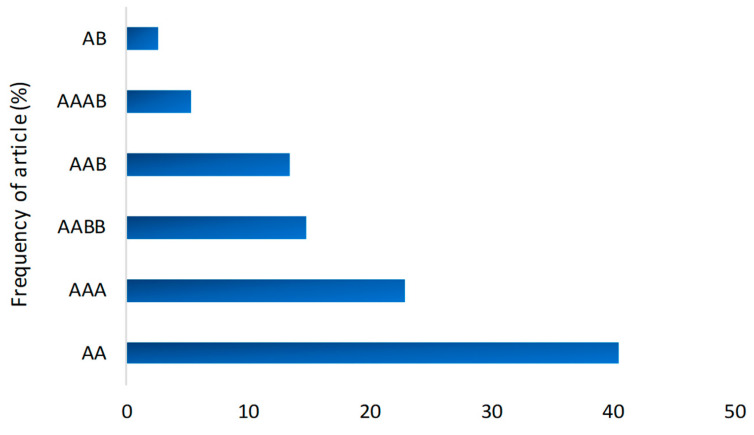
Bar graph of the frequency of genomes according to the resistance/tolerance of the banana genotypes studied.

**Table 1 plants-13-01299-t001:** Scale to assess banana reaction to lesion nematodes according to Pinochet [38].

Plant Response	Root Lesion Index (%)	Corm Grade
Immune	0	0
Resistance	<10	<1
Tolerant	10–20	1–2
Susceptible	20–40	2–4
Highly susceptible	>40	>4

**Table 2 plants-13-01299-t002:** Methods for extracting nematodes from *Musa* spp. roots described in articles published in the last 13 years.

Root Extration Method	Article	Nematode
Macerated roots in a blender followed by sieving	[17,40]	*Radopholus similis*
	[41]	*Radopholus similis*, *Helicotylenchus* spp., *Meloidogyne* spp., *Pratilenchus* spp.
	[49]	*Pratylenchus* spp., *Helicotylenchus* spp., *Meloidogyne* spp., *Radopholus* spp.
	[21,39]	*Helicotylenchus multicinctus*
	[27]	*Pratilenchus coffeae*
Counting of females and juveniles after staining the roots with acid lactophenol fuchsin	[29,45,46,47]	*Meloidogyne incognita*
Manual dissection of the lesioned roots	[50]	*Radopholus similis*
Roots in polypropylene bags submerged in 1% H_2_O_2_	[44]	*Radopholus similis* and mixed population (*R. similis*, *H. multicinctus*, *M. incognita*)
	[43]	*Radopholus similis*
Macerated roots in a blender and extraction using the Baermann technique	[23,42]	*Radopholus similis* and mixed population (*H. multicinctus*; *Meloidogyne* spp.)
	[51]	*Radopholus similis*, *Helicotylenchus multicinctus*, *Meloidogyne* spp.
Sifting and centrifugation with sucrose solution	[52]	*Radopholus* spp., *Helicotylenchus* spp., *Meloidogyne* spp.
Modified Baermann method and staining with acid fuchsin	[53]	*Meloidogyne javanica*, *Rotylenchulus reniformis*, *Helicotylenchus* spp.; *Pratylenchus* spp.
Maceration, flotation and centrifugation technique. And Maceration in sodium hypochlorite (NaOCL), flotation and centrifugation technique	[54]	*Helicotylenchus multicinctus*, *Meloidogyne* spp., *Pratylenchus goodeyi*, *Radopholus similis*
Modified Baermann technique	[55]	*Radopholus similis*, *Pratylenchus coffeae*, *Meloidogyne incognita*, *Rotylenchus reniformis*, *Helicotylenchus dihystera* and others
	[56]	*Radopholus similis*, *Pratylenchus* spp., *Helicotylenchus multicinctus*
	[57]	*Pratylenchus coffeae*
	[58]	*Helicotylenchus multicintus*, *Pratylenchus goodeyi*, *Radopholus similis*, *Meloidogyne* spp.
	[59]	*Radopholus similis*

**Table 3 plants-13-01299-t003:** Resistance of banana genotypes to parasitic nematodes characterized in articles on genetic improvement carried out in the last 13 years.

Musa Germoplasm	Musa Genome	Level of Tolerance or Resistence	Nematode	Article
Yangambi Km5	AAA	PR	*Radopholus similis*	[28]
Yangambi Km5	AAA	R	*Radopholus similis*	[17,22,29,31,40,41,42,50,60,61,62,63]
Yangambi Km5	AAA	T	*Pratylenchus coffeae*	[64]
Yangambi Km5	AAA	T	*Meloidogyne incognita*	[29]
Pisang Lilin	AA	R	*Radopholus similis*	[17,22,31,40,65]
Pisang Lilin	AA	R	*Pratylenchus coffeae*	[27,55]
Pisang Lilin	AA	R	*Helicotylenchus multicinctus*	[21,39]
Pisang Lilin	AA	R	*Meloidogyne incognita*	[45,46,47]
Ro Im V4 6-1-1	AAA	R	*Radopholus similis, Pratylenchus coffeae*	[55]
Si Im V4 10-5-3	AAA	R	*Radopholus similis, Pratylenchus coffeae*	[55]
Anaikomban	AA	T	*Radopholus similis, Pratylenchus coffeae*	[55]
Anaikomban	AA	R	*Pratylenchus coffeae*	[64]
Anaikomban	AA	R	*Radopholus similis*	[17,22,40]
FB920	AAA	T	*Radopholus similis, Pratylenchus coffeae*	[66]
MA13	AAA	T	*Radopholus similis, Pratylenchus coffeae*	[66]
Pisang Jari Buaya	AA	R	*Radopholus similis*	[17,40,41]
Pisang Jari Buaya	AA	R	*Helicotylenchus spp.*	[41]
FHIA-23	AAA	R	*Radopholus similis*	[41]
Valery	AAA	R	*Helicotylenchus spp.*	[41]
Manoranjitham	AAA	R	*Radopholus similis*	[17,40]
Rose	AA	R	*Radopholus similis*	[17,22,40,67]
Matti	AA	R	*Radopholus similis*	[17,40]
Hatitat				
Pisang Mas	AB	T	*Radopholus similis*	[17,40]
Veneettu Kunnan	AB	R	*Radopholus similis*	[17,40]
Then Kunnan	AB	T	*Radopholus similis*	[17,40]
Gros Michel	AAA	T	*Radopholus similis*	[17,40]
Williams				
Red Banana (Mutant)				
Green Red				
Agneeswar				
4279-06	AA	R	*Radopholus similis*	[28]
0323-03				
0337-02				
4249-05	AA	HR	*Radopholus similis*	[28]
Pisang Pipit	AA	PR	*Radopholus similis*	[28]
5854-03				
1318-01				
4285-02				
N118				
Tjau Lagada				
Calcutá 4				
1319-01				
Pa Rayong				
Birmanie				
Vitória				
Thap Maeo				
4223-06				
Pisang Jaran				
Pisang Nangka	AAAB	PR	*Radopholus similis*	[28]
FHIA-21	AAAB	R	*Radopholus similis*	[68]
Kluai Pa 26	AA	R	*Radopholus similis*	[61,69]
K. Nang Nuan	AAB	R	*Radopholus similis*	[61,69]
Pisang Papan	AAA	R	*Radopholus similis*	[61,69]
Tongat	AA	R	*Radopholus similis*	[22]
Tongat	AA	T	*Radopholus similis*	[17,40]
TMB2x 9128-3	AA	R	*Radopholus similis*	[62]
Karthobiumtham	AAB	R	*Pratylenchus coffeae*	[25,36,67,70,71,72,73]
Long Tavoy	AA	R	*Radopholus similis*	[50]
Saba	AAB	R	*Radopholus similis*	[50]
Prata-Anã	AAB	MR	*Meloidogyne javanica*	[74]
BRS Princesa	AAAB	MR	*Meloidogyne javanica*	[74]
BRS Princesa	AAAB	R	*Radopholus similis*	[26]
BRS Japira	AAAB	R	*Radopholus similis*	[26]
BRS Platina				
Latundan	AAB	PR	*Radopholus similis*	[69]
4349-05	_	R	*Radopholus similis*	[63]
H-11-08	_	R	*Radopholus similis*	[22]
H-11-21				
H-11-23				
H-11-25				
H-11-36				
H-11-69				
H-11- 70				
H-11-71				
H-11-76				
H-11-02	_	T	*Radopholus similis*	[22]
H-11-03				
H-11-06				
H-11-12				
H-11-18				
H-11-24				
H-11-37				
H-11-49				
H-11-65				
H-11-78				
H201				
H912	_	R	*Radophous similis*	[42]
H914				
H916				
H926				
H943				
H 903	_	T	*Radophous similis*	[42]
H 906				
H 913				
H 915				
H923				
H934				
H939				
H 904	_	T	*Radophous similis*	[42]
			*Meloidogyne incognita*	[29]
H 911		T	*Radophous similis*	[42]
			*Meloidogyne incognita*	[29]
H 952		T	*Radophous similis*	[42]
			*Meloidogyne incognita*	[29]
H 921	_	T	*Meloidogyne incognita*	[29]
H 924				
H 926				
H 943				
H516	AAA	R	*Meloidogyne incognita*	[45]
			*Pratylenchus coffeae*	[27]
			*Helicotylenchus multicinctus*	[21]
			*Radophous similis*	[75]
H531	AAB	R	*Meloidogyne incognita*	[45]
			*Pratylenchus coffeae*	[27]
			*Helicotylenchus multicinctus*	[21]
			*Radophous similis*	[75]
H511	AABB	T	*Meloidogyne incognita*	[45]
			*Pratylenchus coffeae*	[27]
			*Helicotylenchus multicinctus*	[21]
			*Radophous similis*	[75]
H534	AAB	T	*Meloidogyne incognita*	[45]
			*Pratylenchus coffeae*	[27]
			*Helicotylenchus multicinctus*	[21]
			*Radophous similis*	[75]
H537	AABB	T	*Meloidogyne incognita*	[45]
			*Pratylenchus coffeae*	[27]
			*Helicotylenchus multicinctus*	[21]
			*Radophous similis*	[75]
H571	AABB	T	*Meloidogyne incognita*	[45]
			*Pratylenchus coffeae*	[27]
			*Helicotylenchus multicinctus*	[21]
			*Radophous similis*	[75]
H572	AAB	T	*Meloidogyne incognita*	[45]
			*Pratylenchus coffeae*	[27]
			*Helicotylenchus multicinctus*	[21]
			*Radophous similis*	[75]
H589	AABB	T	*Meloidogyne incognita*	[45]
			*Pratylenchus coffeae*	[27]
			*Helicotylenchus multicinctus*	[21]
			*Radophous similis*	[75]
H-02-34	AABB	T	*Meloidogyne incognita*	[46]
				[47]
H-02-34	AABB	T	*Helicotylenchus multicinctus*	[39]
H-02-34	AABB	T	*Radophous similis*	[75]
H-03-05	AABB	T	*Meloidogyne incognita*	[46]
				[47]
H-03-05	AABB	T	*Helicotylenchus multicinctus*	[39]
H-03-05	AABB	T	*Radophous similis*	[75]
H-03-13	AABB	T	*Meloidogyne incognita*	[46]
				[47]
H-03-13	AABB	T	*Helicotylenchus multicinctus*	[39]
H-03-13	AABB	T	*Radophous similis*	[75]
H-03-17	AABB	T	*Meloidogyne incognita*	[46]
				[47]
H-03-17	AABB	T	*Helicotylenchus multicinctus*	[39]
H-03-17	AABB	T	*Radophous similis*	[75]
H 04-12	AABB	T	*Meloidogyne incognita*	[46]
				[47]
H 04-12	AABB	T	*Helicotylenchus multicinctus*	[39]
H 04-12	AABB	T	*Radophous similis*	[75]
H- 04-24	AABB	T	*Meloidogyne incognita*	[46]
				[47]
H- 04-24	AABB	T	*Helicotylenchus multicinctus*	[39]
H- 04-24	AABB	T	*Radophous similis*	[75]
NPH-02-01	AAB	T	*Meloidogyne incognita*	[46]
				[47]
NPH-02-01	AAB	T	*Helicotylenchus multicinctus*	[39]
NPH-02-01	AAB	T	*Radophous similis*	[75]
H 510	AABB	T	*Helicotylenchus multicinctus*	[39]
			*Meloidogyne incognita*	[47]

HR, R, PR, MR e T: highly resistant, resistant, partially resistant, moderately resistant and tolerant.

**Table 4 plants-13-01299-t004:** Gene expression studies of banana plants infected with nematodes in articles, carried out over the last 13 years.

Tested Genotype	Genes	Sequences of Specific Primer		Nematode	Article
		Forward Primer (5′-3′)	Reverse Primer (5′-3′)		
Karthombiumtham and Nendran	AY427192.1	TGATGTGTGGAATGAGAACGA	CAAGAGCCAGCAATGTTCAA	*Pratylenchus coffeae*	[67]
	AM931368.1	CGTGGAGAGGCTTACCAAAG	GCCAACCATTTCTGCAATCT		
	AM931420.1	CCTGGAGAGCCTTACGAAAG	GTACTGCGGACCTCAATGGT		
	AM931401.1	CCTGGACAGGCTTACCATAC	AACCATGTCGGCAATCTTTC		
	AF227002.1	CAAGAGCCAGCAATGTTCAA	GCAGTGATTTGCAAGCCTTA		
	AM931390.1	CGTCGGGAGGCTAACCAAAG	CCTGGTTCTCCGTACCTCAA		
Karthobiumtham; Nendran; FHIA-1; Anaikkomban; Kunnan; Pisang Jaribuaya; Pisang lilin; Calcutta-4; Yankambi KM-5; Rasthali	poly phenol oxidase (PPO)	GACCGCATGTGGTACTTGTG	GGATCTCGACGTCTTGGTA	*Pratylenchus coffeae*	[70]
Karthobiumtham and Nendran	Metallothionein	GGTCAACTCTGAGACCTGA	CCGAGGTACAGGTA GAACAT	*Pratylenchus coffeae*	[25]
	1,3 Glucanase	GGATGAGACTCTACGATCC	GCCTGATCAAGTTCTGGTTG		
	Chitinase	AGTCAAGGTGATGCTCTCCATC	TCCGGCGATGTTGAAGTCTATG		
	Lipoxygenase	TCCACCAGCTCATCAACCAC	TCAGCAGCTTGAAGATGGGG		
	Cytochrome p450	AGAGCGACTCACAGACTCGAC	CCGGGCAGGTACTTGTAGG		
	Peroxidase	TATGCTCACCATTGCTGCTC	TGATTACCATTGCGAGGACA		
	25S rRNA	ACATTGTCAGGTGGGGAGTT	CCTTTTGTTCCACACGA GATT		
Karthobiumtham and Nendran	WRKY52	TAAGGCGAAGAGGAAGGTGA	TCTCCTGTGTGCATCGGTAG	*Pratylenchus coffeae*	[72]
	WRKY92	AAAGCATCAACCCAGCAAAC	ACGGTGCATCGATAATAGGC		
	WRKY69	GAACCGGATCTGGATCTCAA	CGTTCTTCCCTTCCTCATCA		
	WRKY19	CCAGCTGAATGATCTGACGA	TTGCAATCCTGTCTGACTCG		
	WRKY41	ACGCGAATGTTAGCGTCAAT	CGTGAAGGAAGGAACGATGT		
	WRKY81	AGACAATCCATGCCCAAGAG	TGACTTGAGGTCAGGTGCTG		
*Musa acuminata* 4297-06 and Grad Naine	CALS7	CACCCAGAACATGGTATACTTGAAA	GGTCTCAGGCCTCGTCTTTATG	*Meloidogyne incognita*	[76]
	EXPB11	TAGCAGCAGGAAGTCCTTCGA	GTCGTTCGTCGTGCACAGAA		
	ARR18	CGGATGACGACTCTAGATGCAA	TCGGAGAGGAACACGGAAAA		
	FBXL13	TGGAGTACCTCGGCAAGTTTG	GATGAGATCGTCCTCGCTGATAC		
	EXPA26	CACCTGGGTGCCGATGAC	AAGGCTCTGCCCCACCAT		
	TIFY6B	CAACCGATAGAGTCATCCCTGC	AGTGATCGCTTCATCGAGAGCT		
	ERF4	CCCAAATGTTGGTCCGTTTC	TCGCTGTCTTCCACGATTCA		
	BETV1J	CAGCACTACCATTCGGCTACG	CGAAGAGGGTCTGCTTGCAC		
	APS1	AAGGTCAAGAAGATTGATAGGATATGTG	GTCTTCTGGGAGGTGACAACAAG		
	PER68	CCAAGAAACCACGTAGCAATCA	CAAAATGTGTATGACGTTGGATTCA		

**Table 5 plants-13-01299-t005:** PICOS terms for the research “question” used in the systematic review of genetic improvement methods for nematode resistance in banana cultivation over the last 13 years.

Description	Abbrevion	Question Components
Population	P	Phytoparasitic nematodes of banana plants (*Musa* spp.)
Interest/Intervention	I	Genetic improvement strategies for nematode resistance
Comparison	C	Cultural control methods and chemical or other management strategies
Outcome	O	Tolerance or resistance of banana plants to phytoparasitic Nematodes
Study design	S	Scientific articles

**Table 6 plants-13-01299-t006:** List of questions on genetic improvement of *Musa* spp. for resistance to plant-parasitic nematodes, which will be addressed through a systematic review of studies conducted in the last 13 years.

Research Questions
Q1: What are the main nematode species that affect banana and plantain crops?
Q2: Which cultivars are recognized as resistant to nematodes?
Q3: Which banana breeding programs are focused on nematode resistance or cross-breeding for the purpose of developing resistant cultivars?
Q4: Are there any known sources of nematode resistance?
Q5: Which genes have been reported to be related to nematode resistance?
Q6: How is germplasm selected?
Q7: What are the most used methodologies for extracting nematodes from roots?
Q8: What are the methods for assessing symptoms?
Q9: What existing tools are used to characterize nematode-resistant plants? Are there any molecular markers?
Q10: Are there any studies on the topics of gene editing, cisgenics, and transgenics?
Q11: How often is the banana genome used?

## Data Availability

No new data were created or analyzed in this study.

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
