# Peer review of "Phytoparasitic Nematodes of Musa spp. with Emphasis on Sources of Genetic Resistance: A Systematic Review"

_plants, 2024, doi:10.3390/plants13101299_

Round 1
Reviewer 1 Report
Comments and Suggestions for Authors
The main objective of the review was to assess the current state of knowledge on genetic resistance of Musa spp. to plant parasitic nematodes to aid in their management. However, majority of the review describes the article screening process for the review, collaboration network among countries based on co-authorships, summary of journals the articles published in, frequency of articles by country, frequency of nematodes studied and their distribution in the countries, environment in which the studies were conducted, methods for extracting nematodes, etc. To enhance the significance of the work, I suggest emphasizing the content aligned with the objective within the review.
Figure 9: A word cloud may not be the most suitable choice for scientific visualization. While words stand out by size based on their frequency of occurrence, the length of the word, decorative fonts and different colors used in the figure can mislead readers interpretation. I suggest removing Figure 9.
Comments on the Quality of English Language
Minor edits required.
Author Response
Comments and Suggestions for Authors
The main objective of the review was to assess the current state of knowledge on genetic resistance of Musa spp. to plant parasitic nematodes to aid in their management. However, majority of the review describes the article screening process for the review, collaboration network among countries based on co-authorships, summary of journals the articles published in, frequency of articles by country, frequency of nematodes studied and their distribution in the countries, environment in which the studies were conducted, methods for extracting nematodes, etc. To enhance the significance of the work, I suggest emphasizing the content aligned with the objective within the review.
A: Dear reviewer, we thank you for your valuable contributions to our work. The Systematic Review (SR) has a defined methodology that is already used in our research group, as can be seen in the following publications:
https://doi.org/10.3389/fpls.2021.657916
https://doi.org/10.3390/jof7040249
https://doi.org/10.3390/agronomy13030730
https://doi.org/10.3390/plants12020305.
A: In this sense, the indications of methods for selecting articles, network of interactions between countries and authors, nematode species, among other topics, are part of SRs and have the potential to help other research groups working with banana genetic breeding for resistance to nematodes. In addition, the article presents tables with the profile of resistance to nematodes of different banana genotypes (Table 3) and gene expression (Table 4), which were presented for the first time in a systematic way in our SR. Thus, we sought to aggregate several relevant information from the selected articles in order to contribute to the genetic improvement of bananas for resistance to phytoparasitic nematodes. Therefore, in order for the content to align with the theme of the work, the title was changed following the suggestion of the second reviewer.
Figure 9: A word cloud may not be the most suitable choice for scientific visualization. While words stand out by size based on their frequency of occurrence, the length of the word, decorative fonts and different colors used in the figure can mislead reader´s interpretation. I suggest removing Figure 9.
A: Dear reviewer, thank you for the corrections and suggestions. The word cloud is a type of analysis that is widely used in systematic reviews of the literature. This representation facilitates the identification and visualization of the most relevant sources of resistance to nematodes in a didactic and clear fashion.
Reviewer 2 Report
Comments and Suggestions for Authors
This is a very good manuscript useful for those involved in doing or support research in breeding bananas for resistance to plant parasitic nematodes. In doing so, the authors did a lot of work and selected only the most significant literature. The data reported in most of the figs are useful especially for policy maker to decide if or not and where to invest money for research on bananas. The most interesting part for researches is the information in the tables 3 and 4 listing different sources of resistance and their caracterization.
However, hereafter are suggestions for improving the manuscript or at least making it more readable amd less wordy.
Title: Actually the authors do not present just data on resistance and, therefore, I suggest to change the title as follow: "Phytoparasitic nematodes of Musa spp. with emphasis on sources of genetic resistance: a systematic review"
Summary:
Line 11: is "and one of the main causes are", suggested "of main concern being"
Line 14: is "lose", suggested "reduce"
Line 18: is "breeding methods", should be "sources of resistance to nematodes"
Lines 23-24: delete the last sentence
Other suggestions
line 34: do you mean non fumigant nematocides?
line 44: the parenthesis before 14 must be a square one.
Line 53: is "depends", suggested "relys" or "based"
Table 1 and elsewhere: you report the scale of Pinochet J. as oriental scale. In my opinion this is not correct. Pinochet worked on bananas in Central America, as t can be seen by its different articles. Just say Pinochet scale.
In the opinion of this referee, the methods used to extract nematodes from roots were not the best. A comment by the authors on this aspect is suggested. However, the list in Table 2 is more than exaustive. Therefore, of all these data would be sufficient to mention the principles of the extraction methods and not specify by different authors.
line 380-381: From "conducted" to end of the sentence, I suggest to chamge to " conducted under controlled conditions, involved more than one nematode (n=7)"
line 400 and the folloing: in bracket must be reported the amount of roots/soil to which they refer to.
The discussion must be strongly reduced. The first part appears as an appendix of the results whose details are not of very interest for scientists (that should already know) nor for others. This referee considers not important the lines 324-488. The content of these line have nothing to do with resistance.
The lines 626-629 can be deleted. The statement made is a general one and refers to all crop plants.
Comments on the Quality of English Language
The English quality is very good and just minor editing is suggested
Author Response
Comments and Suggestions for Authors
This is a very good manuscript useful for those involved in doing or support research in breeding bananas for resistance to plant parasitic nematodes. In doing so, the authors did a lot of work and selected only the most significant literature. The data reported in most of the figs are useful especially for policy maker to decide if or not and where to invest money for research on bananas. The most interesting part for researches is the information in the tables 3 and 4 listing different sources of resistance and their caracterization.
However, hereafter are suggestions for improving the manuscript or at least making it more readable amd less wordy.
Title: Actually the authors do not present just data on resistance and, therefore, I suggest to change the title as follow: "Phytoparasitic nematodes of Musa spp. with emphasis on sources of genetic resistance: a systematic review"
A: Text corrected accordingly.
Summary:
Line 11: is "and one of the main causes are", suggested "of main concern being"
A: Text corrected accordingly.
Line 14: is "lose", suggested "reduce"
A: Text corrected accordingly.
Line 18: is "breeding methods", should be "sources of resistance to nematodes"
A: Text corrected accordingly.
Lines 23-24: delete the last sentence
A: Text corrected accordingly.
Other suggestions
line 34: do you mean non fumigant nematocides?
A: This term was used in line 334 and was deleted following the reviewer's later suggestions line 44: the parenthesis before 14 must be a square one.
Line 53: is "depends", suggested "relys" or "based"
A: Text corrected accordingly.
Table 1 and elsewhere: you report the scale of Pinochet J. as oriental scale. In my opinion this is not correct. Pinochet worked on bananas in Central America, as t can be seen by its different articles. Just say Pinochet scale.
A: The alteration was carried out in the Table and throughout the text.
In the opinion of this referee, the methods used to extract nematodes from roots were not the best. A comment by the authors on this aspect is suggested. However, the list in Table 2 is more than exaustive. Therefore, of all these data would be sufficient to mention the principles of the extraction methods and not specify by different authors.
A: Dear reviewer, we thank you for your contributions and corrections to our manuscript. Changes have been made. Table 2 has been reduced to only the extraction principles that were reported in the selected articles that specified the methods. Methods that were indicated by different authors were excluded. We also decided to exclude the counting methods because they were repetitive in the table, and most of them used a microscope. The use of the microscope was included in the text and a comment on the efficacy of the extraction methods was included in the discussion.
line 380-381: From "conducted" to end of the sentence, I suggest to chamge to " conducted under controlled conditions, involved more than one nematode (n=7)"
A: Text corrected accordingly.
line 400 and the folloing: in bracket must be reported the amount of roots/soil to which they refer to.
A: This paragraph was excluded following previous suggestions of the reviewer.
The discussion must be strongly reduced. The first part appears as an appendix of the results whose details are not of very interest for scientists (that should already know) nor for others. This referee considers not important the lines 324-488. The content of these line have nothing to do with resistance.
A: Dear reviewer, thank you for your contributions. In view of their statements, we understand that some topics of the discussion could be summarized and we did so, but that some topics are essential for the discussion in a systematic review, such as the number of studies found and the method of selecting articles adopted. These discussions are important to understand how certain evaluations are conducted for the characterization of nematodes in a breeding program. The topics, "Selection of studies" and "Countries of origin" have been reduced to just one topic.
The lines 626-629 can be deleted. The statement made is a general one and refers to all crop plants.
A: Text corrected accordingly.
Round 2
Reviewer 2 Report
Comments and Suggestions for Authors
In this last corrected version of the manuscript I can see that the authors have considered my suggestions and, therefore, I suggest the publication of the manuscript as it is.